# Targeting neovascularization and respiration of tumor grafts grown on chick embryo chorioallantoic membranes

Hyrije Ademi [1,2‡], Dheeraj A. Shinde[1‡], Max Gassmann[1,3], Daniela Gerst[1], Hassan Chaachouay[4,5], Johannes Vogel[1‡], Thomas A. Gorr[1‡]*

**1** Institute of Veterinary Physiology, Vetsuisse Faculty, University of Zurich, Zurich, Switzerland, **2** Center for Clinical Studies at the Vetsuisse Faculty of the University of Zurich, Zurich, Switzerland, **3** Zurich Centre for Integrative Human Physiology (ZIHP), University of Zurich, Zurich, Switzerland, **4** Division of Radiation Oncology, Vetsuisse Faculty, University of Zurich, Zurich, Switzerland, **5** Bioactives, Health & Environment Laboratory, Epigenetics, Health & Environment Unit, Faculty of Science and Techniques, Moulay Ismail University, Errachidia, Morocco

‡ These authors share first authorship on this work. JV and TAG are joint senior authors on this work.
* tgorr@access.uzh.ch

**Data Availability Statement:** All relevant data are within the manuscript and its Supporting information files.

## Abstract

Since growing tumors stimulate angiogenesis, via vascular endothelial growth factor (VEGF), angiogenesis inhibitors (AIs, blockers of the VEGF signaling pathway) have been introduced to cancer therapy. However, AIs often yielded only modest and short-lived gains in cancer patients and more invasive tumor phenotypes in animal models. Combining anti-VEGF strategies with lactate uptake blockers may boost both efficacy and safety of AIs. We assessed this hypothesis by using the *ex ovo* chorioallantoic membrane (CAM) assay. We show that AI-based monotherapy (Avastin®, AVA) increases tumor hypoxia in human CAM cancer cell xenografts and cell spread in human as well as canine CAM cancer cell xeno-grafts. In contrast, combining AVA treatment with lactate importer MCT1 inhibitors (α-cyano-4-hydroxycinnamic acid (CHC) or AZD3965 (AZD)) reduced both tumor growth and cell dissemination of human and canine explants. Moreover, combining AVA+AZD diminished blood perfusion and tumor hypoxia in human explants. Thus, the *ex ovo* CAM assay as an easy, fast and cheap experimental setup is useful for pre-clinical cancer research. Moreover, as an animal-free experimental setup the CAM assay can reduce the high number of laboratory animals used in pre-clinical cancer research.

## Introduction

Angiogenesis, the formation of new blood vessels from pre-existing ones, is typically dormant in adults. Yet, active angiogenesis represents a hallmark of many pathological progressions, including diabetic retinopathy, rheumatoid arthritis, cardiac ischemia, psoriasis or tumor growth and spread [1]. New capillaries are required for growth of the primary nodule beyond constraints (i.e. size of ~1–2 mm$^3$) [2]. However, the non-productive angiogenesis seen in solid malignancies notoriously yields tortuous, dilated vessels with highly erratic blood flow

**Funding:** TAG: Krebsliga Zürich and Marie-Louise von Muralt-Stiftung für Kleintiere HA: Swisslife MG: Stiftung für wissenschaftliche Forschung an der Universität Zürich The funders had no role in study design, data collection and analysis, decision to publish, or preparation of the manuscript.

**Competing interests:** Additionally, we want to state that although parts of our research were funded by Swisslife, this does not alter our adherence to PLOS ONE policies on sharing data and materials.

**Abbreviations:** AI, angiogenesis inhibitor; CA IX, carbonic anhydrase 9; CAM, chorioallantoic membrane; CHC, α-cyano-4-hydroxycinnamic acid; ddPCR, digital droplet PCR; Flk1, fetal liver kinase 1; Flt1, fms related tyrosine kinase 1; HIF, hypoxia inducible factor; KDR, kinase insert domain-containing receptor; MCT, monocarboxylate transporter; VEGF, vascular endothelial growth factor; VEGFR, vascular endothelial growth factor receptor.

and leaky endothelial lining [3], which is not necessarily associated with an increasing oxygenation of the neoplasm [4]. In fact, even in areas of active angiogenesis a significant portion of tumor tissue is perfused by deoxygenated blood [5]. As a result, there is a considerable spatial and temporal heterogeneity in oxygen partial pressure ($pO_2$) within the tumor. Consequently, a highly proliferative cohort of actively respiring cells can be found in oxygenated areas, while strongly glycolytic or even metabolically depressed and S-phase quiescent cells are nestled at hypoxic, perinecrotic layers [6]. This alteration of metabolic activities, along with maximal vessel-to-cell diffusion distances, renders cells in the deoxygenated areas prone to develop resistance against blood-born cytotoxic drugs and irradiation [7]. The highly heterogeneous composition of tumors with differing drug susceptibilities of individual cell cohorts needs to be considered for more effective anticancer therapies [8].

Tumor hypoxia represents a potent stimulus for continued induction of several key angiogenic cytokines, including the vascular endothelial growth factor (VEGF) [9, 10] and its endothelial-specific receptors VEGFR1 (Flt1) and VEGFR2 (KDR/Flk1) [11, 12]. Stimulated expression of VEGF and VEGFR1/R2 during low $pO_2$ is, in part, conferred via the hypoxia inducible transcription factors 1 and 2 (HIF-1/-2). VEGF binding to VEGFR1, VEGFR2 and VEGFR3 represents a major signaling cascade to promote proliferation and differentiation of the endothelial lineage in response to hypoxia [13] and marks an essential prerequisite of metastasis (cf. above). Given the pivotal importance of VEGF and its receptor VEGFR2 in angiogenesis, initial hopes were high to successfully combat cancer by inhibiting this pathway [14–17]. However, mounting experience from clinical trials to date reveals the sobering fact that VEGF-targeted therapy often prolongs overall survival of cancer patients by only months and does not offer an enduring cure [18–20] or even failed entirely to reveal any evidence regarding improved quality of life [20–22]. These clinical shortcomings of AI-based therapies frequently arise from the development of resistance of the neoplasm to a given compound via the recruitment of alternative signaling modes. Moreover, preclinical studies [14, 23–25] demonstrated, in contrast to others [26, 27], enhanced metastasis in tumor-bearing mice treated with Sunitinib or Avastin®. To produce robust therapeutic gains, combinations of antiangiogenic strategies with measures for the inhibition of metastasis have repeatedly been suggested since sole AI therapies are prone to raise the tumor´s aggressiveness by increasing tissue hypoxia and tolerance of tumor cells to withstand it [14, 20].

Therefore, advancing therapies need to target not only the oxygenated/aerobe compartment of the tumor adjacent to vascular structures via blood-born cytotoxins or AIs but also the vessel-remote severely hypoxic areas. The α-cyano-4-hydroxycinnamic acid (CHC)-mediated block of cellular lactate import via monocarboxylate transporter 1 (MCT1) was shown by Sonveaux et al. to drive oxygenated tumor cells into aerobic glycolysis (Warburg effect) by using glucose instead of lactate as primary energy substrate. According to the authors' metabolic symbiosis concept, oxygenated cells start to increasingly compete with hypoxic areas for glucose, which will eventually kill particularly hypoxic cells prior to their development of therapeutic resistance and a highly metastatic phenotype [28]. Thus, combining CHC or AZD3965, the next generation MCT1 inhibitor, with AIs (to starve oxygenated regions from nutrients), might render hard-to-treat malignancies more susceptible to therapy and is conceptually a very promising anticancer approach.

By applying this treatment strategy, we aimed to evaluate the applicability of the *ex ovo* chorioallantoic membrane (CAM) assay as an animal-free experimental setup in an anti-cancer approach. This assay complies with the 3R principles since according to Swiss legislation (Animal Protection Ordinance, Art 112) it is not considered an animal experiment when terminated prior to embryonic day 14 as until then the CAM tissue remains free of pain-perceiving fibers. The CAM develops during avian ontogenesis and serves as embryonic gas exchange

organ. Because of its prominent vascularization it is used to study angiogenesis and angiogenesis inhibitors [29–31]. Moreover, the early chick embryo is immunodeficient, which allows explantation and growth of cells and biopsies from different species without graft rejection in CAM assays [32, 33]. In the *ex ovo* application, tumor grafts are directly accessible to manipulations (i.e. topical or i.v. drug application) and data collection obtained by optical methods (i.e. Laser speckle imaging). Hence, it is possible to observe and analyze vessel formation and inhibition upon specific treatment with compounds upon explantation of cells or tissue or even visualization with nanoparticles [30–32, 34]. The experimental time window between explantation of the tumor cells and final measurements is 6–7 days. Thus, the assay is particularly useful for rapid screens and safety measure assessments regarding angiogenesis, tumor growth and cell dissemination of fast growing and spreading entities of cancer cells [35].

In our model tumor grafts, angiogenesis and the MCT1-driven lactate uptake were targeted through application of Avastin® (AVA) and CHC or AZD3965 (next generation inhibitor), respectively. Since tissue hypoxia is prevalent in both human [28] and canine solid malignancies [36–39] and the occurrence of metabolic symbiosis among oxic/hypoxic compartments of the tumors was either demonstrated (human model tumors) [28] or can be anticipated to exist (canine tumors), we assessed the efficacy of this combinatorial approach by grafting human glioblastoma (U87), highly invasive breast carcinoma (MDA-MB231) as well as canine oral melanoma (17CM98) and canine osteosarcoma (D17) cells onto the CAM surface. With this setup, we were able to demonstrate 1) the tumor hypoxia and cell spread-promoting effect of Avastin® monotherapy as proof of principle, 2) the superiority of the combinatorial Avastin®/CHC/AZD-based (AVA+CHC or AVA+AZD) therapy for effective targeting of oxygenated/aerobe and hypoxic/glycolytic compartments of the tumor in inhibiting growth of primary masses and cell dissemination and 3) the suitability of the CAM/explant approach as a pre-clinical alternative for low cost, time saving and animal-free experimental setup.

## Results

### Selection of cell lines for tumor grafts

Tumor growth, solid mass formation and angiogenesis were scored after 6 days of inoculation (d9-d14) for 13 different cell lines (Table 1). Human glioblastoma U87 (Fig 1), canine 17CM98 and canine D17 cell explants fulfilled the scoring criteria tumor growth (assessed as visible increase in tumor mass; see Fig 2) together with active neovascularization (assessed as increased blood flux around tumor ("halo"); see Fig 3) best and, therefore, were selected for all subsequent experiments. Since *in vivo* glioblastomas scarcely metastasize outside the central nervous system [40, 41] U87 explants were used as a rapidly growing, highly angiogenic tumor model to analyze treatment-related changes in proliferation, perfusion and tissue oxygenation, whereas tumor cell dissemination was assessed using MDA-MB231 breast cancer cells due to their known strong invasive and metastatic potential in various settings [42, 43].

### Tumor growth in response to drug treatment

For the final treatment of the CAM explants, we used 10mg/kg AVA and 60mg/kg CHC. These dosages had been found to not exert any toxic effects in preliminary experiments on embryos and cultured U87 cells (see S1 Fig).

Fig 2a, demonstrates on exemplary U87 grafts the tumor growth kinetics in response to six days of AVA- and CHC mono- or combination-therapy compared to vehicle (CTRL). Whereas AVA as well as CHC monotherapy was able to diminish further tumor growth the combination of AVA+CHC therapy induced significant shrinking of U87 model tumors. Data shown in Fig 2b are based on image analysis and demonstrate that, by day 6 of the CAM assay,

**Table 1. Scoring of cell line performance on the CAM.**

| Type | Cell Lines | Proliferation | Solid mass formation | Angiogenesis | Total score |
|---|---|---|---|---|---|
| Human melanoma | 501 Mel | + | - | - | + |
| Human renal carcinoma | RCC4 | - | - | - | - |
| Human renal carcinoma | RCC4/VHL | - | - | - | - |
| Human hepatoma | Hep3B | + | + | + | +++ |
| Human breast carcinoma | MCF7 | + | + | + | +++ |
| Human breast carcinoma | MDA-MB468 | + | - | + | ++ |
| Human breast carcinoma | MDA-MB231 | + | - | - | + |
| Human glioblastoma | U87 | + | + | + | +++ |
| Canine oral melanoma | 17CM98 | + | + | + | +++ |
| Canine osteosarcoma | D17 | + | + | + | +++ |
| Canine soft tissue sarcoma | K9STS | - | - | - | - |
| Canine hemangiosarcoma | HAS | - | - | - | - |
| Murine melanoma | B16F10 | + | - | + | ++ |

Performance of tumor grafts from 13 different cell lines on the CAM. A positive score was given, for proliferation (observed growth: +; dried mass of dead cells: -), solid mass formation (explant on CAM remains as dome-shaped coherent nodule: +; explant on CAM diffuses away from site of inoculation: -), and angiogenesis (observed formation of capillary network as angiogenic „halo"seen in Laser-Doppler imaging (Fig 3): +; no increased flux: -). CAM suitability was found to be maximal for Hep3B, MCF7, U87, 17CM98 and D17 cells, of which the latter three were selected for further analyses.

U87 CTRL grafts increased their volume by 269% ± 16.4%. In contrast, AVA and CHC treated U87 tumors grew until d6 by 139% ± 11.89% and 122% ± 22.28%, respectively. In contrast, AVA+CHC treated tumor grafts shrunk to 50% ± 41.13%. Thus, AVA and CHC monotherapy significantly inhibited the growth of primary masses, while combining AVA+CHC appears to have an additive effect, resulting in significant tumor shrinkage.

Canine explants were treated with AVA in the same dosage as human explants (10mg/kg) but with AZD3965 (2.5µM/egg), a next generation MCT1 inhibitor, that compared to CHC is of higher specificity. Unfortunately, quantification of canine tumor growth required a different experimental approach, since none of the canine tumors used grew as spherical masses as human U87 explants did. Therefore, quantification was performed by qPCR using specific primers against canine, human and chicken hypervariable D-loop region of mitochondrial DNA. In human U87 grafts this measurement of tumor growth showed similar results compared with image analysis (no further growth increase and even a slight tumor growth reduction, with overlapping error bars relative to CTRL tumors) for AVA treatment, the AVA +AZD combinatorial intervention reduced growth to 46.96±10.45% if the CTRL tumor mass was set to 100% (Fig 2c). Canine 17CM98 tumors responded to AVA treatment with a growth reduction of 60.03±8.8%, to AZD treatment with 57.36±3.9% and to the combination of AVA +AZD with 37.59±6.2% (Fig 2d). D17 explants responded to AVA with a 36.49±5.78% growth reduction, to AZD with 56.74±15.64% and to AVA+AZD with 32.68±2.17% (Fig 2e). Thus, AVA and CHC or AZD monotherapies stalled or slightly reduced growth of primary masses, while the combination of AVA with CHC or AZD potentiated shrinkage of the primary mass in human as well as canine model tumors.

## Tumor perfusion

Assessment of perfusion was performed using Laser speckle contrast imaging. In order to separate the CAM perfusion from that of the yolk beneath the CAM a polystyrene plastic strip

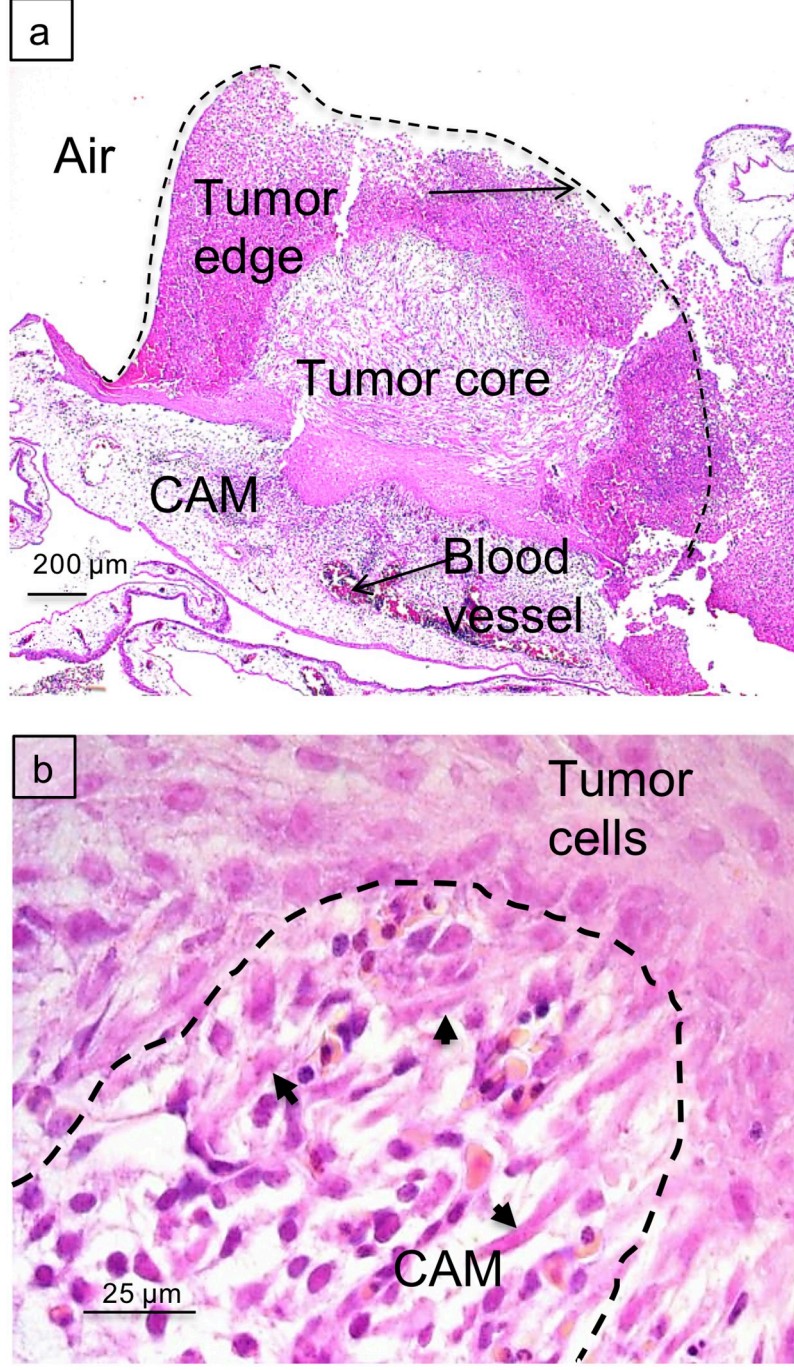

**Fig 1. Hematoxylin/Eosin staining of paraffin sections of U87 tumor grafts 6 days after inoculation. a**: overview of explant/CAM interface (scale bar: 200μm). **b**: detailed view showing tumor cells (arrows) invading CAM parenchyma with hatched line indicating invasion front (scale bar: 25μm).

was inserted underneath the explant site. With this setup perfusion around and on the tumor could be determined as a function of AVA (10mg/kg), CHC (60mg/kg) and AVA+CHC treatment (Fig 3a and 3b) in human U87 model tumors as well as in canine 17CM98 and D17 explants treated with either AVA (10mg/kg), AZD (2.5μM/egg) or AVA+AZD (Fig 3c–3f).

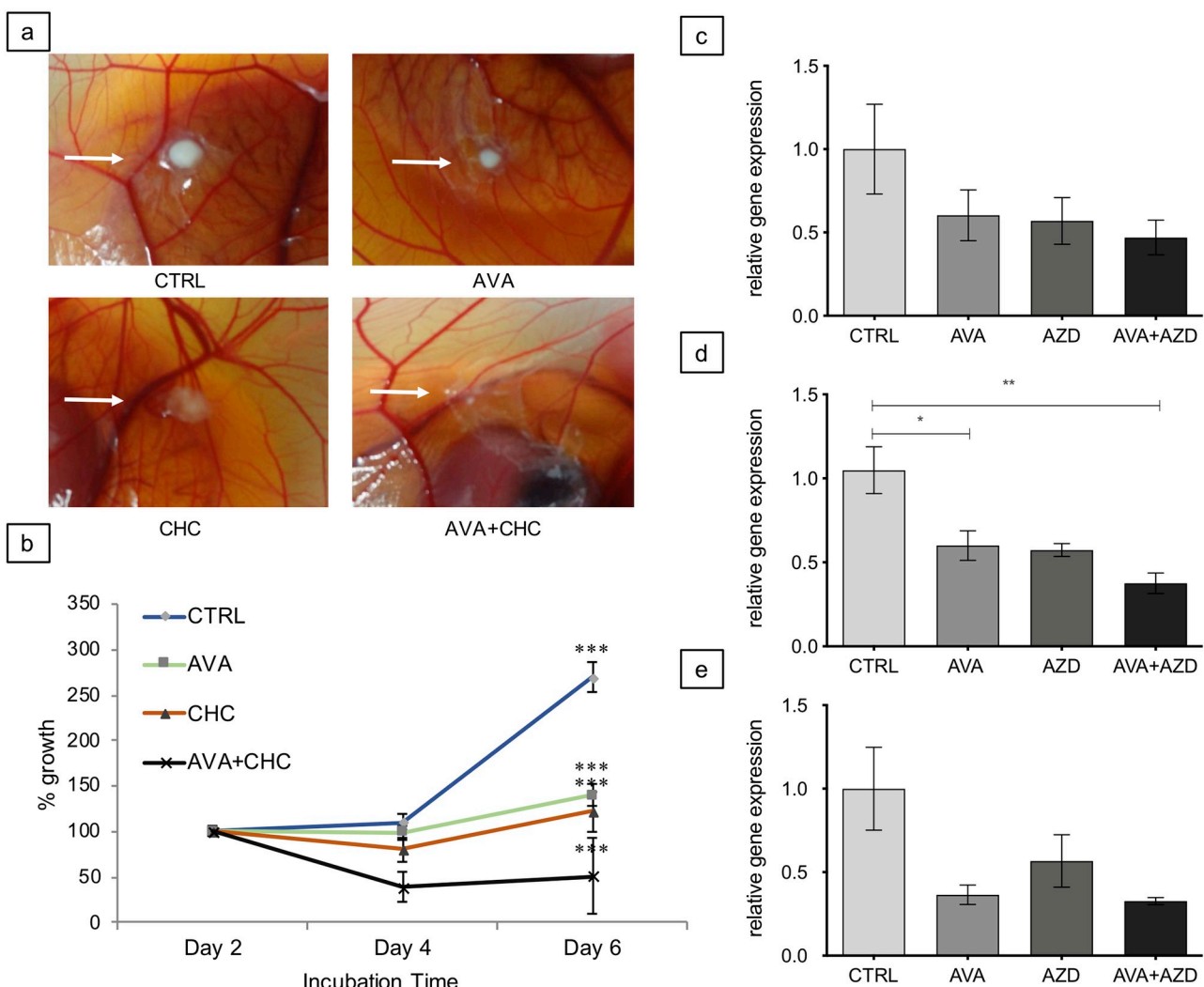

**Fig 2. Tumor growth as function of drug treatment. a**: Representative human U87 tumor explants subjected to treatments as indicated (from top left to bottom right): CTRL (PBS, 10µl topically applied around explant); Avastin® (AVA; i.v. injected: 10mg/kg); α-cyano-4-hydroxycinnamic acid (CHC, 60mg/kg; topically applied around explant); AVA+CHC (10mg/kg+60mg/kg). White arrows point to respective primary mass. **b**: Quantification of tumor growth kinetics of human U87 by image analysis between day 2 (considered 100%) and 6 of CAM inoculation Day 0 = explant day. Data are means ± SEM, n = 6. Statistical differences were tested with two-way ANOVA. *** = p<0.001. **c + d + e**: Growth quantification using D-loop PCR data obtained with human specific (c) and canine specific (d +e) primers normalized to D-loop amplicon abundance of chicken host tissue, as a function of treatment (AVA, AZD and AVA+ AZD), relative to control (CTRL) conditions. Tumor explants (c: human U87; d: canine 17CM98 and e: canine D17) were treated as indicated: control (PBS, 50µl i.v. injected, CTRL); Avastin® (i.v. injected, 10mg/kg, AVA); AZD3965 (i.v. injected, 2.5µM/egg, AZD); AVA+AZD (10mg/kg + 2.5µM/egg). Data in c), d) and e) are means ± SEM, n = 6. Statistical differences: one-way ANOVA, * = p < 0.05, ** = <0.02.

Most grafts reproducibly displayed a "halo" of ongoing angiogenesis around the tumor. For human explants the size of the 'halo' was visibly reduced in parallel with the therapeutically induced growth arrest or shrinkage of the tumor (Fig 2a and 2b). Fig 3b illustrates the continuous reduction in blood flow at the tumor periphery when comparing CTRL treatment with AVA or CHC monotherapies and AVA+CHC combination, respectively. Again, the AVA +CHC combinatorial treatment was most effective in diminishing perfusion to the explant to background levels. Of note, perfusion at remote CAM control sites and embryonic angiogenesis was not affected by any treatment.

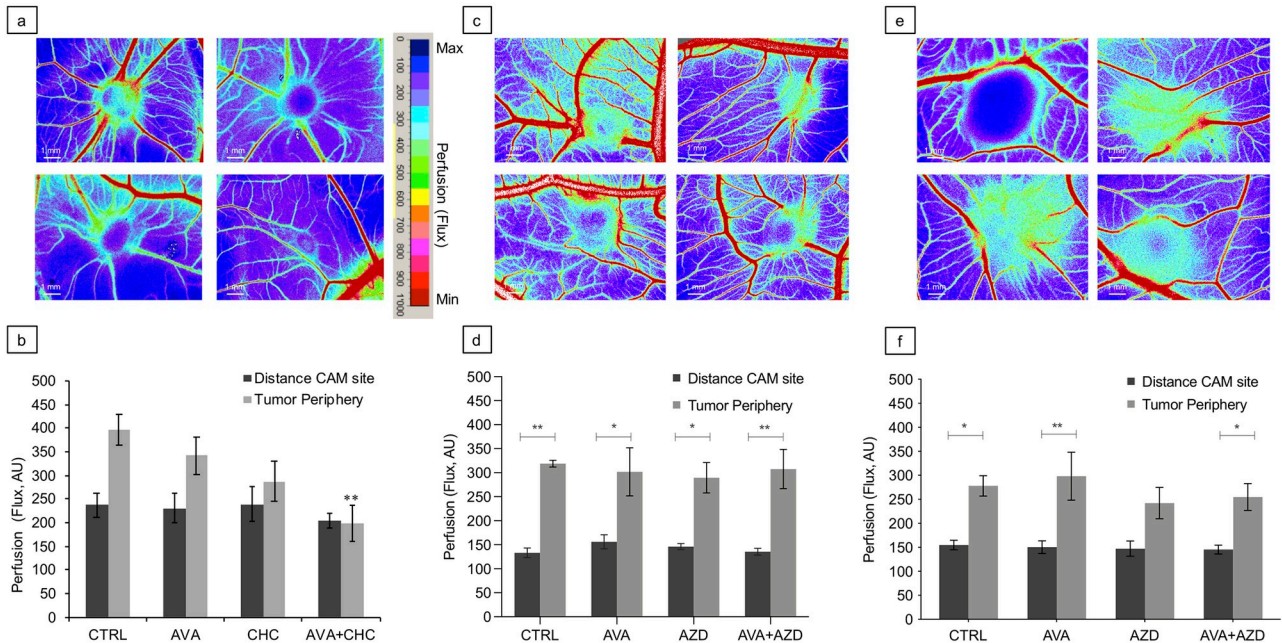

**Fig 3. Blood perfusion of tumor explants. a + c + e**: Representative perfusion images of tumor explants (a): U87; (c) 17CM98 and (e): D17, subjected to the indicated treatments. Minimal flow = blue; maximal flow = red; see color scale on the right of panel a. Note the greenish ring ("halo") of angiogenic activity around the tumor grafts. **b + d +f**: Quantified flux of the tumor vicinity (b): U87; (d): 17CM98 and (f) D17, in response to indicated treatments. Data are means ± SEM, (b) n = 6, (d) n = 5; (f) n = 8. Statistical differences were tested with two-way ANOVA. * = p < 0.05, ** = <0.02. Scale bars = 1mm. AU = arbitrary units.

In canine explants, however, the various treatments were ineffective in reducing perfusion into the mass (Fig 3c–3f) even though an increase in blood flow in the periphery of explants ('halo') was reproducibly detected.

## Assessment of tumor hypoxia

Local tissue hypoxia in response to AVA, CHC and AVA+CHC treatment of human U87 explants was assessed through staining of i.v. injected pimonidazole (cf. Fig 4a: brightfield and 4b: pimonidazole staining). Pimonidazole signals (red) quantified as ratio of maximal pimonidazole intensity to average background intensity (Fig 4c) served as proxy for the severity of tissue hypoxia, and % of pimonidazole-positive area to total tumor area (Fig 4d) as a measure for the extent of tissue hypoxia. In spherical (dome-shaped) U87 grafts, pimonidazole staining was absent in regions close to the air-exposed surface and for most of the highly vascularized CAM/tumor interface (cf. Fig 1). In contrast, pimonidazole accumulated mainly within the tumor's core. Importantly, AVA monotherapy significantly augmented intensity (+44% of CTRL explants, Fig 4c), and moderately also area (+18% of CTRL explants, Fig 4d), of pimonidazole staining in U87 tumors, which provides strong evidence for an Avastin®-dependent increase in tissue hypoxia in these experimental tumor grafts. In contrast, CHC-only and AVA +CHC reduced pimonidazole intensities to 90% (CHC) and 61% (AVA+CHC), and areas to 69% (CHC) and 74% (AVA+CHC), of CTRL explants, respectively. In line with the metabolic symbiont concept [28], CHC-based formulations yielded a selective kill of hypoxic, pimonidazole-positive cells. With regards to explants of canine cancer cells, pilot experiments revealed an intensified pimonidazole staining signal of cell cultures exposed to 16hrs hypoxia (0.1% $O_2$) when compared to normoxic cultures (16hrs, 21% $O_2$). Unfortunately, the staining of canine

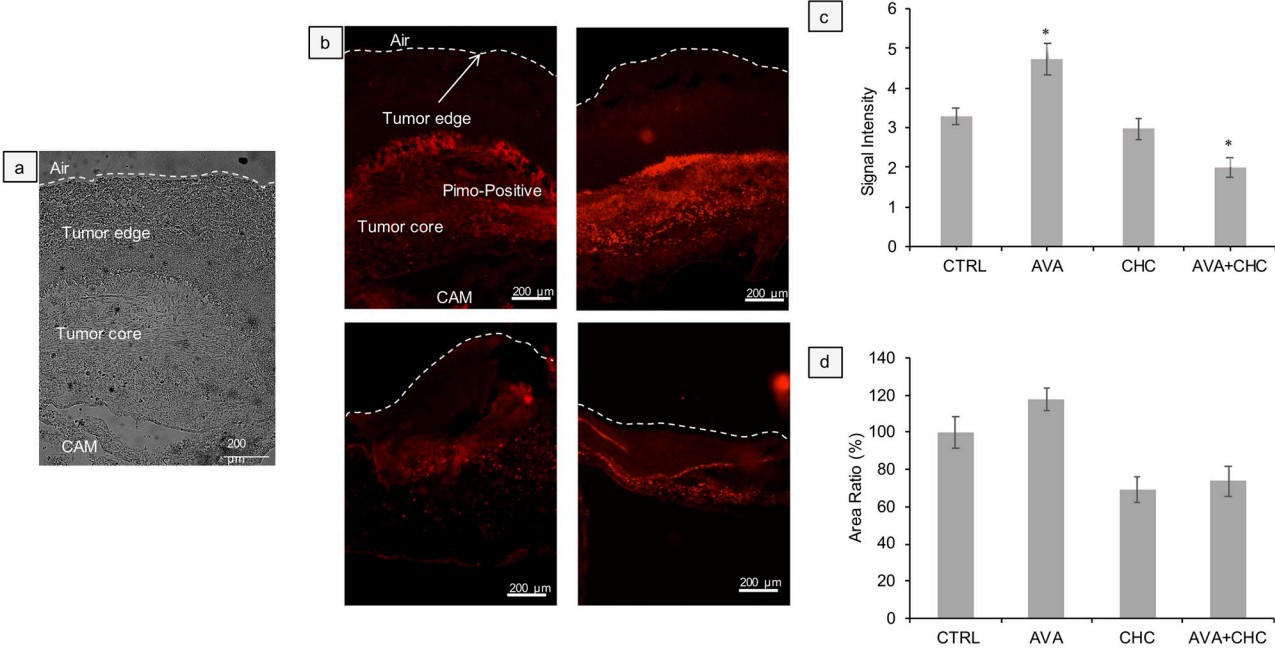

**Fig 4. Assessment of local tumor hypoxia. a**: Representative brightfield image for illustration of different tumor regions. **b**: Representative images of pimonidazole (Pimo) stained (red) cross sections of U87 tumors after CTRL (PBS), AVA, CHC and AVA+CHC treatments. Note the designated tumor regions in the upper left Pimo-stained panel. Both the severity of tissue hypoxia (**c**), indicated by the intensity ratio of pimonidazole signals (maximal Pimo signal / average tissue background) and the extent of tissue hypoxia (**d**), as depicted by the percent area ratio (pimo-positive area / total tumor area), were quantified as function of treatment. Data are means ± SEM, n = 4. Statistical differences were tested with one-way ANOVA. $^*$ = p<0.05.

xenografts with pimonidazole did not reveal visible treatment-dependent changes of tissue hypoxia in 17CM98 and D17 grafts (S2 Fig). Similar results, unable to discern among treatments, were obtained with carbonic anhydrase IX (CA IX) immune-histochemistry (S3 Fig).

## Spreading of tumor cells

Using quantitative droplet digital PCR (ddPCR) in conjunction with zonal DNA extraction (Fig 5a) and human- or chicken-specific mitochondrial DNA D-loop primer/probe pairs (Fig 5b and S1 Table) enabled us to quantify invasion of the highly aggressive MDA-MB231 cells from the primary inoculation site into the surrounding CAM tissue as a function of treatment. Regarding human cancer cells we had to switch from the non-metastasizing U87 to the actively disseminating and invasive MDA-MB231 cells. The human/chicken D-loop signal ratio of the tumor itself (zone 0) was set to 100%. Spreading of cells to zone 1 (directly adjacent to the explant) resulted in a ratio of the human/chicken D-loop signal of 30.68±3.2% in CTRL, 56.35 ±15.49% in AVA, 22.49±1.1% in CHC and 14.97±1.9% in AVA+CHC respectively. This indicates maximal cell motility because of AVA application. A similar observation was made for zone 2 (CTRL 1.8±0.4%, AVA 6.0±0.1%, CHC+CHC 1.5±0.7% and AVA+CHC 0.5±0.4%), while negligible amounts of human DNA were detected in zone 3. Overall, these data indicate that AVA treatment increases dissemination of MDA-MB231 cells as compared to CTRL and other treatments. In contrast, and in comparison, to CTRL, cell spread was reduced with CHC and even more with AVA+CHC treatment (Fig 5c). Regarding tumor grafts of canine cells, the qPCR technique was used. Compared to control, we could show for both types of canine explants an increased spreading tendency to zone 1 after AVA monotherapy. Although relative DNA expression was quite low for 17CM98 grafts, spreading of D17 cells into zone 1, always

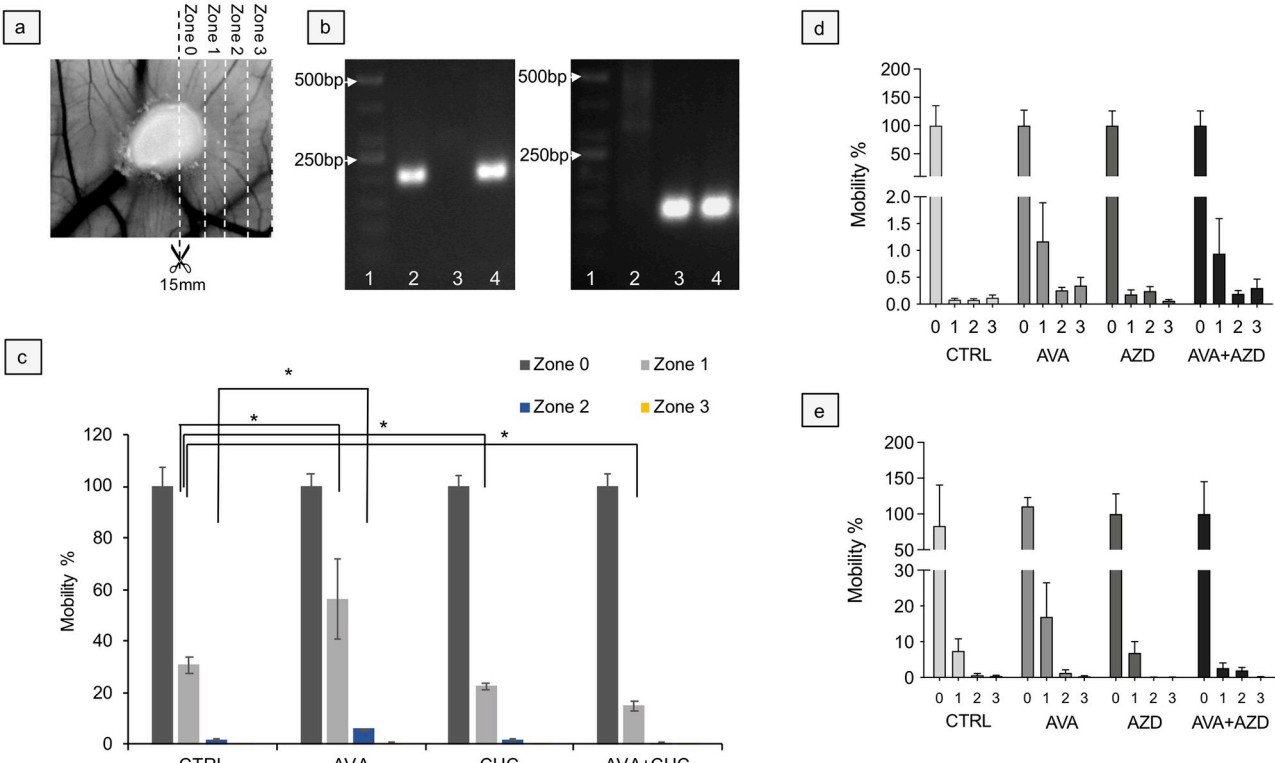

**Fig 5. Dissemination of tumor cells in response to treatment. a**: For quantification of cell dissemination genomic DNA (gDNA) and mitochondrial DNA (mtDNA) were extracted from MDA-MB231, 17CM98 and D17 tumor explants and nearby CAM tissue, divided into 4 zones (Zone 0, 1, 2, and 3; 2mm x 15mm stripes as indicated). **b**: Using human or chicken specific PCR primers against the hypervariable D-loop region of mtDNA (see S1 Table for sequences and fluorescent labels) pilot PCRs revealed species-specific amplification of the D-loop amplicon in comparison with positive control gDNA from human MDA-MB231 cells and chicken liver. Left gel: PCR using human specific D-loop primers; right gel: PCR using chicken specific D-loop primers. Loading of either gel: lane 1 = DNA Mw marker; lane 2 = DNA extraction (gDNA+mtDNA) from cultured MDA-MB231 cells; lane 3 = DNA extraction from chicken liver; lane 4 = DNA extraction from CAM-resident MDA-MB231 explant. **c**: Content of human DNA, normalized to content of chicken (CAM derived) DNA, was quantified for zones 0–4 of MDA-MB231 explants using ddPCR with human or chicken specific D-loop primers/probes, to estimate cell motility as function of treatment. For each treatment, zone 0 (explant periphery) was set to 100% of detected MDA-MB231 DNA signals. Data are means ± SEM, n = 3. Statistical differences were tested with Wilcoxon rank-sum test. * = p<0.05. d + e: Spreading behavior of 17CM98 and D17 cells, respectively, were quantified by qPCR with specific primers against mitochondrial D-Loop of canine (signal) and chicken (normalization) sequences (S2 Table). Handling of the samples was exactly as described in (a). **d**: canine 17CM98 explants showed increased spreading into zone 1 in response to AVA monotherapy and in the combination therapy. **e**: canine D17 showed overall a higher spreading compared to canine 17CM98. These explants demonstrated the superiority of the combination therapy in diminishing the spreading tendency of cells compared with CTRL and AVA-monotherapy. Data are means ± SEM, d) n = 12 and e) n = 6. Statistical analysis: Kruskal-Wallis test, ns.

compared to zone 0, was 7.4±3.4% in CTRL, 16.92 ± 9.5% in AVA, 6.9 ± 0.3% in AZD and 2.6 ± 0.8% in AVA+AZD treated grafts. Similar to an aggressive human carcinoma cell, canine explants showed a maximal dissemination tendency in response to AVA alone, while AZD and even more AVA+AZD appeared to diminish cell dissemination from the primary mass, an observation that was more obvious in D17 cell grafts (Fig 5d and 5e).

## Metabolic symbiosis

Since the lactate exporting (MCT4) and importing (MCT1) transport proteins are crucially important for the metabolic symbiont concept, expression of these proteins was assessed at mRNA level in U87 human tumor explants using qPCR. By trend, expression of MCT1 and MCT4 in U87 tumor grafts was higher after AVA monotherapy (Fig 6). Unfortunately, neither the MCT1 nor the MCT4 transcript could be detected in canine tumor grafts. We, therefore

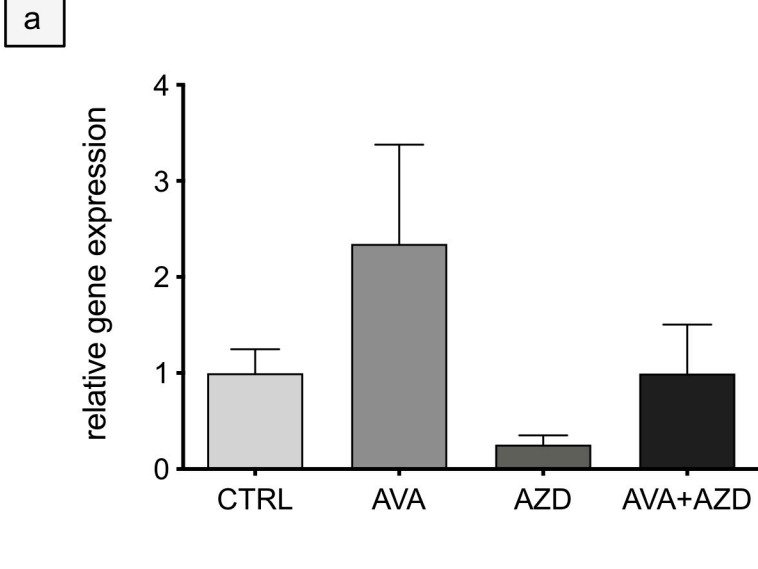

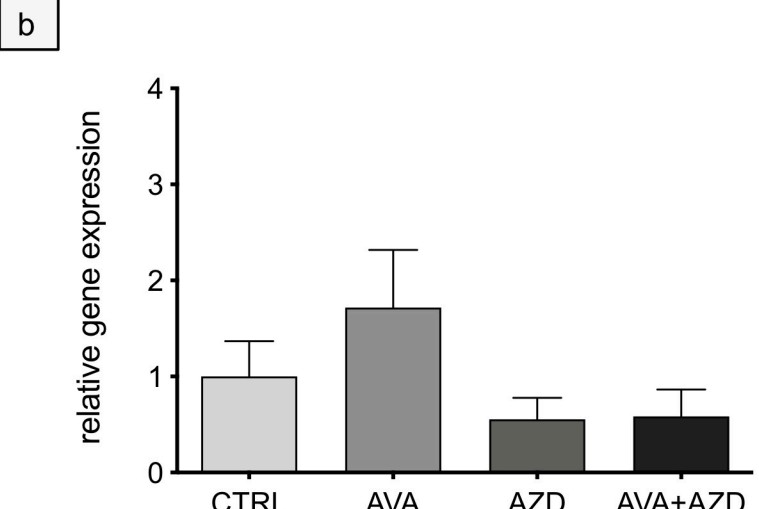

**Fig 6. Quantitative PCR with human-specific primers against MCT1 and MCT4 as a function of treatment (AVA, AZD, AVA+AZD) of human U87 tumor explants.** MCT1 (**a**) and MCT4 (**b**) gene expression was normalized to β-actin abundance and relative to control (CTRL). Data are means ±SEM, n = 4. Statistical analysis: Kruskal-Wallis test, ns.

used the highly sensitive *in situ* hybridization technique (RNAscope® ISH) with specific, manufacturer-generated probes against canine MCT1 and MCT4 in cell cultures. With this staining procedure, an increase of MCT1 and MCT4 expression was clearly visible in hypoxic (1% $O_2$, 16h) compared to normoxic cell cultures (S4 and S5 Figs).

## Discussion

The current study aimed to evaluate the applicability of the *ex ovo* chorioallantoic membrane (CAM) assay as an animal-free experimental setup by targeting, individually and simultaneously, both aerobic and hypoxic compartments of model tumors through anti-VEGF and lactate import blocking anti-MCT1 intervention strategies, respectively.

Using human U87 glioblastoma cells as xenografts (Table 1), solid tumor masses on the CAM demonstrated, within a 6-day period of inoculation (d9-d14), increase in tumor volume (Fig 2), active angiogenesis (Fig 3), development of local tissue hypoxia (Fig 4) and moderate cell invasion into the chorioallantoic mesenchyme (Fig 5c). These points document that the model reliably mirrors key features of growth and behavior of high-grade glioma *in vivo*. Regarding canine 17CM98 oral melanoma and D17 osteosarcoma cells as xenografts we demonstrated solid CAM masses, reduced biomass accumulation by any treatment (Fig 2d and 2e), active angiogenesis (indicated by 'halo'; Fig 3) and a moderately intensified cell dissemination tendency in response to AVA monotherapy. In addition to techniques previously applied in CAM studies [44–46], we employed the following novel methodological adaptations to determine the treatment´s impact on the following read-outs: i) real time perfusion in angiogenic CAM tumor model by Laser-Speckle contrast imaging; ii) severity and extent of local tissue hypoxia in CAM xenografts by i.v. injection of pimonidazole; iii) cell dissemination by zonal DNA extraction in conjunction with human or canine specific qPCR. Growth, vascular functional performance and tissue oxygenation of U87, 17CM98 and D17 grafts was recorded in response to separate and combined application of non-toxic doses of the anti-VEGF antibody Bevacizumab (Avastin®, AVA) and α-cyano-4-hydroxycinnamic acid (CHC) or AZD3965 (AZD), respectively. CHC inhibits the lactate importer MCT1 with roughly 10-fold selectivity over other MCTs [28], whereas AZD is a specific MCT1 inhibitor [47]. By virtue of this anti-VEGF/anti-MCT1 combinatorial strategy we anticipated to antagonize not only the oxygenated/aerobe compartment of the tumor mass via AVA but also to starve the hypoxic cells via CHC or AZD3965. Importantly, this selective kill of cells in the deoxygenated areas through MCT1 inhibition will be accomplished without the need of the often efficacy-limiting delivery of the drug into the hypoxic cells *per se*.

In line with these anticipations, we found the combination of AVA with CHC to be the only mode of intervention successful in shrinking the primary mass of U87 grafts, roughly to half the initial volume, whereas the monotherapies only prevented further mass accumulation of the initial explant (Fig 2). The U87 growth kinetics agreed well with the observed progressive reduction of perfusion (Avastin®+CHC = 51% reduction). Previous CAM experiments with Bevacizumab supported our findings of the antiangiogenic effect, both in CAM and *in vivo* assays [48]. Yet only the combination therapy was able to significantly reduce tumor perfusion down to CAM background levels, thus negating any tumor angiogenesis-related additional flux (Fig 3b). Of note, flux at tumor-remote CAM control sites and, allegedly, embryonic angiogenesis was not affected by any of these interventions (Fig 3b), confirming the non-toxic nature of the compounds and dosages (mono and combinatorial) *in vivo*. The slowing of tumor growth and incomplete block of perfusion by AVA of the human U87 xenografts can be understood as remaining VEGF-independent angiogenesis signaling along with increased tissue hypoxia, which emerged in AVA-treated glioblastoma mouse xenografts [49]. When we used U87 grafts on CAM, we similarly saw that a single intravenous injection of AVA alone increased, not diminished, the intensity (44% increase relative to CTRL explants) and extension (+18% increase) of pimonidazole-positive tumor hypoxia (Fig 4), a finding in line with the known selection towards more hypoxia-tolerant and virulent secondary growth as implicated by the above mentioned Avastin®- or Sunitinib treatments of tumor-bearing mice [14, 23–25]. On the other hand, in U87 grafts and in support of a selective eradication of deoxygenated tumor cells, the Avastin®+CHC combinatorial treatment effectively and significantly diminished pimonidazole-positive hypoxia intensity (39% reduction) and area (26% reduction) (Fig 4). The superior anticancer potency of anti-VEGF/anti-MCT1 combinatorial interventions over respective monotherapies was thus demonstrated by its maximal efficacy in diminishing growth, perfusion and tissue hypoxia within the primary mass of U87 CAM

explants. In concordance with the pimonidazole staining, also the CA IX staining reveled a similar trend of an AVA-induced increase of tissue hypoxia compared with CTRL (S3 Fig).

Unfortunately, U87 grafts did not produce any measurable spread and formation of micro-metastatic foci in distant organs (e.g., liver) within the brief time window at hand. For this reason, cell motility CAM assays were conducted with the strongly disseminating and invasive human breast carcinoma line MDA-MB231 [42, 43].

Spreading of MDA-MB231 cells from the explant to adjacent zones 1 and 2 evidenced significantly elevated cell motility in response to AVA-only application and further illustrates the negative impact an anti-VEGF monotherapy might have. Yet, the data also revealed lesser-than-CTRL cell dissemination after CHC-only and, particularly, after AVA+CHC combinatorial application (Fig 5c). However, the mechanistic basis for this motility-promoting and -inhibiting effect by AVA and CHC-based protocols in MDA-MB231 tumor grafts must fundamentally differ from the metabolic symbiont concept [28] because the lactate importer MCT1 is absent in MDA-MB231 cells due to hypermethylation-mediated gene silencing [28]. Future work thus needs to assess if a possible slowing of MCT4-mediated lactate export in AVA-treated MDA-MB231 grafts translates into higher cellular concentrations of lactate, which might drive the augmented motility and invasiveness seen in Fig 5c. For oxidative (but not Warburg-type) tumors, physiological millimolar lactate levels were found to inhibit the prolylhydroxylase domain 2 (PHD2)-driven hydroxylation and subsequent degradation of HIF-1$\alpha$, thus aiding in the activation of HIF-1 signaling, VEGF secretion and tumor angiogenesis even in normoxic compartments of the malignant mass [50]. These data solidify previously established positive correlations between high-lactate tumors and high incidence of metastasis, increased radiotherapy-resistance and poor clinical outcome [51, 52].

Treatment of the canine explants was conducted by applying AVA or AZD3965 alone and in combination with one another. In agreement with the U87 data, tumor growth measurements in canine explants also demonstrated reduced mass accumulation by both mono- and the combination therapy (Fig 2c and 2d). While perfusion measurements of canine cancer cell explants revealed an intensified perfusion in the vicinity of the grafted tumors compared to remote area, this elevated blood supply proved to be completely refractory to any sort of treatment (Fig 3d and 3f). Although Avastin® is a humanized monoclonal antibody against VEGF, it was shown to cross-react and bind also canine VEGF, with however a marginally reduced affinity compared to human VEGF [53, 54]. We attempted to promote a more prominent effect by applying more AVA (100mg/kg, i.e., 10x the dose used for Fig 3, S6 Fig) and repeated the perfusion measurement. However, this dose not only reduced the blood flow in vicinity of the tumors but also that of the embryonic background. Therefore, higher doses of AVA interfered with the embryonal angiogenesis and would only confound our results on tumor perfusion. On the other hand, the observation on 100mg/kg AVA reconfirms the applied compound to exert no toxic effects on the embryo's vasculature, since background flow was only reduced when higher doses of AVA were applied but no reduction was observed in our final experimental setup with 10mg/kg AVA. Regarding the treatment-refractive nature of the canine explant vasculature, it should be stressed that general drawbacks of CAM assays may well include a) nonspecific inflammatory reactions and b) ongoing embryonic physiological angiogenesis as possible confounding effects on the vasoproliferative responses of the tumor neovascularization. These responses may operate through different and VEGF-independent signaling cascades [31].

Unfortunately, measurements of tumor tissue hypoxia with pimonidazole did also not reveal any clear indication of tissue hypoxia in canine tumor explants (S2b and S2c Fig). Absence of measurable hypoxia induction could be explained by the smaller masses and the more two-dimensional growth pattern of canine tumors in comparison to U87 explants,

which would aid oxygenation of the cancer cells through diffusion from ambient air. Of note, accumulation of pimonidazole-based adducts was well illustrated in cultures of hypoxic canine cancer cells (data not shown). This suggests tissue hypoxia to occur in CAM tumor models of fast-growing masses as observed with U87 glioblastoma explants or at local sites within grafts bulky enough to render diffusion of ambient oxygen insufficient.

Interestingly, the dissemination data showed an overall higher spreading tendency of D17 osteosarcoma compared with 17CM98 oral melanoma. Both cell types displayed an intensified spreading from the primary mass into adjacent zones upon AVA-monotherapy when compared with CTRL explants. Importantly, the combination therapy AVA+AZD reverted this AVA-induced dissemination back to, and for D17 osteosarcoma cells even below, background levels seen in the CTRL group.

Considering that also canine tumors are known to develop severe local hypoxia [55] with increased VEGF plasma levels [38, 39, 56] it is anticipated for canine neoplasms too to represent metabolic symbionts of oxic/hypoxic compartments. Hence, a combination treatment (AVA+AZD) could be beneficial to combat these cancers as well. In order to assess presence of the metabolic symbiosis, the expression of the main transporters (MCT1 and MCT4) was measured in 17CM98 and D17 cell cultures and cell explants. By using RNAscope® *in situ* hybridization on cultured cells an increase of both MCT1 and MCT4 mRNA was seen in 17CM98 oral melanoma cells in response to hypoxic exposure (0.2% $O_2$, 16h) (S4 and S5 Figs). Unfortunately, this expression was not detected in the CAM tumor explants of either canine cell type. Reasons for this failed detection in CAM explants are currently unclear. Possibly the strong oxygenation of the canine tumor grafts by ambient air impacted the expression, since normoxic exposure of cultured 17CM98 and D17 cells also yielded very low levels of expression of either transporter.

Taken together, we were able to evaluate the CAM/tumor assay presented here by simultaneously targeting tumor angiogenesis and hypoxic/glycolytic tumor cells. With this approach we could assess tumor growth, measure blood flow, stain tissue hypoxia and quantify dissemination of human and canine cancer cells. Moreover, we demonstrate that VEGF-targeting treatments, when applied as monotherapy, can increase tumor hypoxia and cell spread, a finding clearly in line with the proposed selection of more hypoxia tolerant and metastasis-prone malignant clonal variants in response to Avastin®- or Sunitinib®-related interventions. The combination of Avastin® with an inhibitor of the lactate trafficking system between hypoxic and oxygenated tumor compartments was able to potently suppress cell dissemination in human or dog cancer cells. Therefore, we demonstrated the suitability of the CAM/tumor approach as a pre-clinical alternative for low cost, time saving and, most importantly, animal-free experimental setup in cancer research.

## Materials and methods

### Reagents

Cell culture media and supplements were purchased from GIBCO Switzerland. The Bevacizumab anti-VEGF antibody Avastin® (herein abbreviated: AVA) was purchased from Roche (Genentech: Cat. # 2918877), α-cyano-4-hydroxycinnamic acid (Cat. # C-2020; herein: CHC) from Sigma-Aldrich and AZD3965 (Cat. # HY-12750; herein: AZD) was purchased from AstraZeneca.

### Cell culture

Human glioblastoma U87 MG (HTB-14™), murine melanoma B16-F10 (CRL-6475™) and canine osteosarcoma D17 (CCL-183™) cell lines were purchased from ATCC® (American

Type Culture Collection). Human breast cancer MDA-MB231 was a generous gift from Prof. E. Dahl (Hospital of the RWTH Aachen University). Canine oral melanoma 17CM98 were kindly provided by Prof. Dr. David Vail, University of Wisconsin-Madison. Canine soft-tissue sarcoma K9STS and hemangiosarcoma HAS were kindly provided by Prof Dr. Carla Rohrer-Bley, Vetsuisse Faculty Zurich. Human hepatoma (Hep3B), melanoma (501), renal clear cell carcinoma with (i.e., RCC4) and without (i.e. VHL reverted cell line: RCC4/VHL) loss-of-function mutation in the VHL tumor suppressor gene as well as breast carcinoma (MCF7, MDA-MB468) cells were available as laboratory stocks from previous works [57, 58]. HEP3B, U87, RCC4 and RCC4/VHL, MCF7, MDA-MB468 cells were cultured in Dulbecco's Modified Eagle's Medium (DMEM), B16F10 cells in Eagle's minimum essential medium (EMEM) and 501 melanoma cells in RPMI medium (RPMI Media 1640). Each medium contained high glucose levels (4.5g/l), supplemented with 10% fetal bovine serum and 1% penicillin/streptomycin. Selection pressure in RCC4 and RCC4/VHL cultures was maintained by addition of G418 antibiotic (0.5mg/ml). Canine cell lines (D17, 17CM98, K9STS and HAS) were cultured in RPMI medium (RPMI Media 1640) supplemented with 10% fetal bovine serum, 1% penicillin/streptomycin, 1% HEPES, 1% sodium pyruvate, 1% non-essential amino acid (NEAA) and 1% GlutaMax. All cell lines were kept at 37˚C in humidified air with 5% $CO_2$ and a $pO_2$ of 141.6 mmHg ([$O_2$] = 18.6% $O_2$).

## *Ex ovo* chick chorioallantoic membrane (CAM) assay

**a) Preparing chick embryos.** Fertilized eggs, (purchased from http://www.animalco.ch), were placed for 72 hours into a cabinet incubator (HEKA, Rietberg, Germany) at 37.5˚C and 60–62% humidity [59]. On day 3 (d3) of chick development, eggs were cracked along the equator, opened and transferred into small, sterile plastic bowls (Thermoflex, Lonay, Switzerland) while ensuring that yolk sac and embryo came to lie facing upwards. These *ex ovo* CAM cultures were covered with Petri dish lids and kept at 37˚C and 60% humidity in Forma Scientific incubators (Model-3336). All experiments were terminated before d14.5 prior to onset of pain sensation in early development, which prevents CAM assays to be considered animal experiments [60].

**b) Selection of a suitable tumor cell line.** On d9, 13 various human, mouse and canine tumor cell lines, individually suspended in 15 μl of medium, were loaded onto the CAM to induce tumor formation. Successful tumor establishment, analyzed by scoring growth, solid mass formation and angiogenesis between d9-d14, resulted in the final selection of human U87 glioblastoma (CAM load: 4 million cells), canine 17CM98 oral melanoma (4 million cells + 1/5 Matrigel) and canine D17 osteosarcoma (6 million cells + 1/5 Matrigel) as lines of interest for further experiments.

## Treatment with Avastin® and CHC or AZD3965

One day after tumor cell inoculation on the CAM, assays were treated with control conditions (i.e., CTRL; PBS (50μl)) or treated with the following experimental agents: a) AVA (10mg/kg; intravenous injection); b) MCT1 inhibitor, either CHC (60mg/kg; topical application around explant) or AZD3965 (2.5μM/egg; intravenous injection) as monotherapy (i.e. CHC; AZD); c) the combination of AVA with either MCT1 inhibitor (i.e. AVA+CHC or AVA+AZD). Prior to these treatments, AVA, CHC and AZD dosages were ascertained to exert no toxicity on cultured cells and embryos (see S1 Fig).

## Quantification of tumor growth

U87 glioma cells were grafted onto the CAM (d9) and tumor growth was monitored by digital imaging of tumors on d10, d12 and d14. Analysis of these images utilized MCID™ 7.0 software (Ontario, Canada), yielding manually traced tumor borders and assumed spherical tumor volume (i.e. volume v = $4/3\pi r^3$, with r = $1/2\sqrt{(D1 \times D2)}$; D1 = short axis, D2 = long axis of tumor area) [61]. The relative volume increase between day 2 (d10, set to 100%) and day 4 (d12) and 6 (d14) of AVA, CHC and AVA+CHC treated tumors was compared with PBS-treated control explants.

Since canine tumor explants did not grow in a spherical fashion as the human U87 cells, quantification of canine cell explants was done by qPCR. In this regard, canine cells were explanted onto the CAM (d9), treated the next day, followed by tumor collection at d14 and the isolation of genomic DNA, containing both nuclear and mitochondrial genomes. In order to distinguish human or canine from chicken amplicons, species-specific primer pairs of the hypervariable D-Loop region of mitochondrial DNA were generated (S2 Table). The abundance of qPCR amplicons was measured in a semi-quantitative manner, with the $\Delta\Delta$CT method [62].

## Quantification of tumor perfusion

Perfusion of the tumor explant and its periphery was measured on d14 using Laser-Speckle perfusion imaging (moorFLPI, Moor Instruments, Germany). Interference by perfusion of the underlying yolk sack was blocked by a polystyrene strip placed underneath the CAM. Perfusion data were recorded, as function of treatment, at high resolution/low speed settings (10sec/frame) in 8 different regions of interest (ROIs, each 4200-pixel large) on and around established tumor grafts on the CAM. For assessing natural blood flow variations of the CAM, four additional ROIs were placed on tumor-remote CAM sites.

## Quantification of tumor hypoxia

The hypoxia marker pimonidazole (aka Hypoxyprobe™-1; NPI Inc., USA) was injected i.v. (60mg/kg) into tumor-bearing embryos on d14, 20 min prior to sacrificing the embryo (i.e. through injection of 100mM KCL) and harvesting the tumor explant. Dissected tumors were fixed in 4% paraformaldehyde for 48h, cryo-protected in 30% sucrose for another 48h, frozen in -20˚C cold isopentane and sliced into 14μm cryo-sections. Next, sections were blocked for 1h with 1% goat serum in PBS and incubated overnight with anti-pimonidazole rabbit IgG (NPI Inc; pAb2627, 1/800 in PBS at 4˚C) and a Cy-3 conjugated secondary goat antirabbit antiserum (Jackson ImmunoResearch, USA, 111.165.003; 1/700 in PBS for 1h at 37˚C). All sections were imaged using identical microscope, camera and acquisition settings to quantify staining intensities and stained area with MCID software. As measure for severity of tissue hypoxia the maximal signal of stained region was normalized to the average background signal of the tissue (intensity ratio). Spatial extent of tumor hypoxia was defined as ratio of the stained ("pimo-positive") area per total tumor area (area ratio). Area ratios obtained from treated tumors are expressed relative to that of control sections, set to 100%.

## Tumor cell dissemination assay

Since U87 cells did not produce any measurable spread to distant organs with the brief experimental time window of the assay, highly invasive MDA-MB231 breast carcinoma [42, 43] cells were used. Thus, 6 days old MDA-MB231 grafts were subsequently sectioned into 2x15mm stripes (Fig 5a). This way four spreading zones were defined with zone 0 representing the

graft's core and zones 1–3 the surrounding CAM at a distance of 2, 4 and 6mm from the tumor periphery. In these zones the concentration of chicken and human DNA was determined using digital droplet PCR (QX100, ddPCR™, Biorad) with chicken and human specific TaqMan primer/probe pairs that were directed against the hypervariable D-loop region of human (i.e., MDA-MB231 cells) or chicken (i.e., amplification of host DNA as loading control) mitochondrial DNA (for sequences, see S1 Table). Motility of MDA-MB231 cells was expressed, for each zone, as percentage of human DNA compared to the chicken DNA.

Regarding canine sequences, however, TaqMan primer/probes designed against the canine mitochondrial D-loop region did not amplify the anticipated PCR product. Therefore, spreading behavior of the cells was quantified using semi quantitative PCR. Accordingly, 6d old 17CM98 and D17 grafts were sectioned as described above and amplicons of canine and chicken of the mitochondrial D-Loop were quantified. Again, data was analyzed using the ΔΔCT [62] method and zone 0 was set to 1. Gene expression of zone 1 to 3 were calculated relatively compared with zone 0.

## Metabolic symbiosis

To gain insight into the metabolic aspect of the treatment specific primers against human MCT1 and MCT4 (28) were used. Reference primers against β-actin were used for amplicon normalization (S2 Table). Expression of the transporters were quantified by qPCR and analyzed using the ΔΔCT method [62].

## Statistical analysis

Results are expressed as mean values ± standard error of the mean of at least 3 independent experiments. Statistical analysis was performed with GraphPad Prism 8 (GraphPad Software, USA). Normal distribution of data population was tested using Shapiro-Wilk test and significance determined by one-way ANOVA, Kruskal-Wallis test, Wilcoxon rank-sum test or two-way ANOVA for comparison between different treatment groups. Tukey or Dunn's post-hoc correction was applied. A p value <0.05 was considered significant.

## Supporting information

**S1 Raw images.**
(PDF)

**S1 Table. Primer/Probe sequences used in ddPCR.**
(PDF)

**S2 Table. Primer used in SYBR green qPCR.**
(PDF)

**S1 Fig. In vitro drug toxicity assay.**
(PDF)

**S2 Fig. Tumor tissue hypoxia on paraffin section—Pimonidazole.**
(PDF)

**S3 Fig. Tumor tissue hypoxia on paraffin section—Carbonic anhydrase IX (CA IX).**
(PDF)

**S4 Fig. Canine MCT4 staining.**
(PDF)

**S5 Fig. Canine MCT1 staining.**
(PDF)

**S6 Fig. Flux measurement on D17 tumor explants treated with 10-times higher AVA concentration.**
(PDF)

## Acknowledgments

JV and TAG are indebted to Dr. Rudolf Steiner, Zurich, for his inspiring introductory CAM assay lab course.

Furthermore, we would like to thank Prof. E. Dahl, Hospital of the RWTH Aachen University, Prof. Dr. David Vail, University of Wisconsin-Madison, and Prof. Dr. Carla Roher-Bley, Vetsuisse Faculty Zurich, for generously providing us with cells utilized in the experiments. We also would like to express our gratitude to Ms. Julia Baumann for a final check and improvement of the manuscripts'language.

## Author Contributions

**Conceptualization:** Hyrije Ademi, Dheeraj A. Shinde, Johannes Vogel, Thomas A. Gorr.

**Funding acquisition:** Hyrije Ademi, Max Gassmann, Johannes Vogel, Thomas A. Gorr.

**Investigation:** Hyrije Ademi, Dheeraj A. Shinde, Daniela Gerst.

**Methodology:** Hyrije Ademi, Dheeraj A. Shinde, Daniela Gerst, Hassan Chaachouay.

**Project administration:** Hyrije Ademi, Johannes Vogel.

**Supervision:** Johannes Vogel, Thomas A. Gorr.

**Writing – original draft:** Hyrije Ademi, Dheeraj A. Shinde, Thomas A. Gorr.

**Writing – review & editing:** Hyrije Ademi, Dheeraj A. Shinde, Max Gassmann, Hassan Chaachouay, Johannes Vogel, Thomas A. Gorr.

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
