## [Decision Letter · Decision Letter 0]

23 Mar 2021

PONE-D-21-03224

Targeting neovascularization and lactate-fueled respiration in model tumors: assessment of anti-cancer treatment efficacy through chick embryo chorioallantoic membrane assays

PLOS ONE

Dear Dr. Gorr,

Thank you for submitting your manuscript to PLOS ONE. After careful consideration, we feel that it has merit but does not fully meet PLOS ONE’s publication criteria as it currently stands. Therefore, we invite you to submit a revised version of the manuscript that addresses the points raised during the review process.

Both reviewers recognize the importance of the paper and the value of the method. However, the editor agrees that the introduction needs to be streamlined and improved, Furthermore, the missing information will have to be added and other minor issues indicated by the reviewers should be corrected.

We look forward to receiving your revised manuscript.

Kind regards,

Filomena de Nigris, M.D., Ph.D.

Academic Editor

PLOS ONE

Journal Requirements:

4.  Thank you for stating the following in the Financial Disclosure section:

"TAG: Krebsliga Zürich and Marie-Louise von Muralt-Stiftung für Kleintiere

HA: Swisslife

MG: Stiftung für wissenschaftliche Forschung an der Universität Zürich

The funders had no role in study design, data collection and analysis, decision to

publish, or preparation of the manuscript"

We note that you received funding from a commercial source: "Swisslife"

5. Please amend either the title on the online submission form (via Edit Submission) or the title in the manuscript so that they are identical.

Reviewers' comments:

Reviewer's Responses to Questions

**Comments to the Author**

1. Is the manuscript technically sound, and do the data support the conclusions?

Reviewer #1: Yes

Reviewer #2: Yes

2. Has the statistical analysis been performed appropriately and rigorously? 

Reviewer #1: Yes

Reviewer #2: Yes

3. Have the authors made all data underlying the findings in their manuscript fully available?

Reviewer #1: Yes

Reviewer #2: Yes

4. Is the manuscript presented in an intelligible fashion and written in standard English?

Reviewer #1: No

Reviewer #2: Yes

5. Review Comments to the Author

Reviewer #1: In this paper Dr. Ademi et al. show their ability to evaluate the ovo chorioallantoic membrane (CAM) assay/tumor assay by simultaneously targeting tumor angiogenesis and hypoxic/glycolytic tumor cells. With this approach they assess tumor growth, measure blood flow, stain tissue hypoxia and quantify dissemination of human and canine cancer cells, demonstrating the suitability of the CAM/tumor approach as a pre-clinical alternative for low cost, time saving and animal-free experimental setup in cancer research.

Data are concordant with recent literature papers as regards the angiogenesis-related assays and animal-free experimental setup in an anti-cancer approach using canine cell lines.

Major Remarks:

• Since the paper is based on the use of CAM authors should explain better this assay as an extraembryonic membrane commonly used in vivo to study both new vessel formation and its inhibition in response to tissues, cells, or soluble factors (i.e. Ribatti D, Gualandris A, et al. J Vasc Res. 1997; Ribatti D, Vacca A et al. Curr Pharm Biotechnol. 2000; Ribatti D, Nico B, et al. Anat Rec. 2001).

Minor Remarks:

• In the Figure 5b, the specific primers used are lacking. In the same Figure, in the plots d and e the figure legends are missing. The authors should indicate this to make the figure understandable.

• In the Figure 6, the figure legends are missing. The authors should add them.

• In the Materials and Methods section the sources and relatives affiliations of several reagents are missing. The authors should complete the informations.

• Please remove the abbreviations that are not necessary and introduce the missing abbreviations.

• All figures layout issues should be resolved in order to submit the paper for publication.

Reviewer #2: In the study” Targeting neovascularization and lactate-fueled respiration in model tumors: assessment of anti-cancer treatment efficacy through chick embryo chorioallantoic membrane assays” the authors report the interesting application of antiangiogenic molecules, demonstrating the results with different types of dynamic experiments, which do not involve animal models. Although the study brings new insights into angiogenesis and its inhibition, it nevertheless needs some modifications.

Minor revision

The text should be revised according to the following points:

- Reduce the "Introduction" paragraph, focusing on the main aspects and the purpose of the study

- Discussion: the part between 304-311 lines should be eliminated or merged with the introduction

- Update the "References" section with more recent citations, if possible

- Insert an "Abbreviations" section

- Figure 3: make the stairs uniform to the same unit

6. PLOS authors have the option to publish the peer review history of their article (what does this mean?). If published, this will include your full peer review and any attached files.

Reviewer #1: No

Reviewer #2: **Yes: **Dr Schiano Concetta

---

## [Author Response · Author response to Decision Letter 0]

30 Apr 2021

Reviewer#1 

1. Since the paper is based on the use of CAM authors should explain better this assay as an extraembryonic membrane commonly used in vivo to study both new vessel formation and its inhibition in response to tissues, cells, or soluble factors (i.e. Ribatti D, Gualandris A, et al. J Vasc Res. 1997; Ribatti D, Vacca A et al. Curr Pharm Biotechnol. 2000; Ribatti D, Nico B, et al. Anat Rec. 2001).

Thank you for the comment. Introduction was revised and rewritten. The papers mentioned have been added to the reference list. 

2. In the Figure 5b, the specific primers used are lacking. In the same Figure, in the plots d and e the figure legends are missing. The authors should indicate this to make the figure understandable.

Primers used in Fig 5b are listed in S1 Table.

We apologize that the legends to Fig 5d and e were not so easy visible. We now highlighted (bold) the letters “d” and “e” in the figure legend text for better visibility of the different panel legends of the figure. 

3. In the Figure 6, the figure legends are missing. The authors should add them.

Maybe the figure legends were overlooked because they were inside the manuscript text. 

4. In the Materials and Methods section the sources and relatives affiliations of several reagents are missing. The authors should complete the informations.

Thank you for this hint. We now added all sources and affiliations of the used reagents 

5. Please remove the abbreviations that are not necessary and introduce the missing abbreviations.

All unnecessary abbreviations were removed, and the missing abbreviations added

6. All figures layout issues should be resolved in order to submit the paper for publication.

All figures were adjusted according to the comments above 

Reviewer#2

1. Reduce the "Introduction" paragraph, focusing on the main aspects and the purpose of the study

Thank your for the comment. Introduction paragraph was shortened and rewritten in order to focus on the main aspects

2. Discussion: the part between 304-311 lines should be eliminated or merged with the introduction

The lines were deleted and included in the introduction

3. Update the "References" section with more recent citations, if possible

Where possible references were updated 

4. Insert an "Abbreviations" section

Thank you for this hint. An “Abbreviations“ section was included after the “Abstract” 

5. Figure 3: make the stairs uniform to the same unit

Figure 3 was adapted according to your suggestions.

---

## [Editor Report · Decision Letter 1]

3 May 2021

Targeting neovascularization and respiration of tumor grafts grown on chick embryo chorioallantoic membranes

PONE-D-21-03224R1

Dear Dr. Gorr,

We’re pleased to inform you that your manuscript has been judged scientifically suitable for publication and will be formally accepted for publication once it meets all outstanding technical requirements.

Kind regards,

Filomena de Nigris, M.D., Ph.D.

Academic Editor

PLOS ONE

Additional Editor Comments (optional):

The revision has addressed the issues previously raised by the reviewers and editor.
---

## [Editor Report · Acceptance letter]

6 May 2021

PONE-D-21-03224R1 

Targeting neovascularization and respiration of tumor grafts grown on chick embryo chorioallantoic membranes 

Dear Dr. Gorr:

I'm pleased to inform you that your manuscript has been deemed suitable for publication in PLOS ONE. Congratulations! Your manuscript is now with our production department. 

Kind regards, 

on behalf of

Prof. Filomena de Nigris 

Academic Editor

PLOS ONE